# AXL receptor tyrosine kinase is required for T cell priming and antiviral immunity

Edward T Schmid[1], Iris K Pang[1†], Eugenio A Carrera Silva[1‡], Lidia Bosurgi[1], Jonathan J Miner[2], Michael S Diamond[3,4,5], Akiko Iwasaki[1,6], Carla V Rothlin[1*]

[1]Department of Immunobiology, School of Medicine, Yale University, New Haven, United States; [2]Department of Medicine, Washington University School of Medicine, St Louis, United States; [3]Department of Molecular Microbiology, Washington University School of Medicine, St Louis, United States; [4]Department of Pathology and Immunology, Washington University School of Medicine, St Louis, United States; [5]The Center for Human Immunology and Immunotherapy Programs, Washington University School of Medicine, St Louis, United States; [6]Howard Hughes Medical Institute, Yale University, New Haven, United States

*For correspondence: carla.rothlin@yale.edu

Present address: †Centre of Influenza Research, School of Public Health, Li ka Shing Faculty of Medicine, The University of Hong Kong, Hong Kong, China; ‡Institute of Experimental Medicine-CONICET, National Academy of Medicine, Buenos Aires, Argentina

Competing interests: The authors declare that no competing interests exist.

**Abstract** The receptor tyrosine kinase (RTK) AXL is induced in response to type I interferons (IFNs) and limits their production through a negative feedback loop. Enhanced production of type I IFNs in $Axl^{-/-}$ dendritic cells (DCs) in vitro have led to speculation that inhibition of AXL would promote antiviral responses. Notwithstanding, type I IFNs also exert potent immunosuppressive functions. Here we demonstrate that ablation of AXL enhances the susceptibility to infection by influenza A virus and West Nile virus. The increased type I IFN response in $Axl^{-/-}$ mice was associated with diminished DC maturation, reduced production of IL-1β, and defective antiviral T cell immunity. Blockade of type I IFN receptor or administration of IL-1β to $Axl^{-/-}$ mice restored the antiviral adaptive response and control of infection. Our results demonstrate that AXL is essential for limiting the immunosuppressive effects of type I IFNs and enabling the induction of protective antiviral adaptive immunity.

## Introduction

AXL is a member of the TAM (TYRO3, AXL, and MERTK) subfamily of RTK that potently inhibits the production of type I IFNs (*Bhattacharyya et al., 2013*; *Rothlin et al., 2015*; *Rothlin et al., 2007*; *Zagorska et al., 2014*). In DCs, AXL is an IFN-stimulated gene (ISG) and hijacks molecular components of type I IFN signaling to induce the expression of Suppressor of Cytokine Signaling (SOCS) 1 and SOCS3 (*Rothlin et al., 2007*). SOCS1 and SOCS3, in turn, downregulate type I IFN signaling. Therefore, AXL is a key component of a homeostatic mechanism that controls type I IFN levels.

Recent studies using an array of enveloped viruses have identified AXL as an enhancer of infection in vitro, including in DCs (*Bhattacharyya et al., 2013*; *Meertens et al., 2012*; *Morizono et al., 2011*; *Shimojima et al., 2007*; *2012*). Enveloped viruses exploit apoptotic mimicry by exposing phosphatidylserine on their lipid envelopes. Binding of phosphatidylserine to the AXL agonist growth arrest-specific 6 (GAS6) protein leads to the activation of AXL on host cells (*Bhattacharyya et al., 2013*; *Lew et al., 2014*). Activation of AXL through viral apoptotic mimicry leads to the induction of the *Socs* genes and the suppression of type I IFN production and signaling (*Bhattacharyya et al., 2013*). It was also shown that the non-enveloped virus SV40 can engage AXL directly by structural mimicry to facilitate infection (*Drayman et al., 2013*). Type I IFNs were identified based on their ability to inhibit the propagation of viruses (*Isaacs and Lindenmann, 1957*; *Taniguchi et al., 1980*). Accordingly, genetic ablation of *Axl* resulted in an enhanced production and

**eLife digest** The immune system must be ever vigilant to ward off infections. Immune cells called T-cells can identify and eliminate microbes, but if they are too aggressive, they can damage the body. To prevent this, the body has systems that control immune responses. For example, another type of immune cell called a dendritic cell produces proteins known as type 1 interferons, which help to fight viral infections while limiting other immune responses.

An enzyme called AXL blocks the production of type 1 interferons. Many scientists believe that this activity reduces the ability of individual cells in the body to defend themselves against attacking viruses. In fact, experiments with cells grown in the laboratory have shown that some viruses activate the AXL enzyme to help them infect. Similar studies have also shown that inhibiting AXL and related enzymes can make cells more able to fight off certain types of viral infection. These and other studies suggested that some drugs that block AXL might be useful treatments for viral infections, however it was not clear if this was the case for all viruses.

Now, Schmid et al. show that the loss of AXL actually makes mice more prone to infections by the influenza virus and West Nile Virus. In the experiments, mice genetically engineered to lack AXL were more likely than normal mice to become ill after exposure to one of the viruses. Furthermore, fewer T cells matured to the stage where they could attack the virus in these mice.

Next, Schmid et al. show that blocking the production of type 1 interferons in the mice that lack AXL restores their ability to fight off these viral infections. This is because type 1 interferons limit the production of a protein that helps the dendritic cells to mature. Therefore, Schmid et al.'s findings show that AXL is vital for mice to fight off viral infections because it helps to balance the antiviral and immune suppressing activities of type 1 interferons. The findings also suggest that using drugs that block AXL to treat infections with certain viruses, including influenza and West Nile Virus, might do more harm than good.

signaling of type I IFN during viral infection of cells in vitro and increased the resistance of DCs to the virus (*Bhattacharyya et al., 2013*). These studies have speculated that disabling AXL RTK function might have potent antiviral activity in vivo (*Bhattacharyya et al., 2013*; *Meertens et al., 2012*; *Morizono et al., 2011*; *Shimojima et al., 2007*; *2012*).

Type I IFNs also mediate a vast array of immunoregulatory functions (*McNab et al., 2015*). For example, sustained production of type I IFNs during chronic lymphocytic choriomeningitis (LCMV) infection inhibited the generation of virus-specific T cells and prevented viral clearance (*Teijaro et al., 2013*; *Wilson et al., 2013*). Similar detrimental effects of type I IFNs have been described during bacterial infections. In particular, type I IFNs inhibit protective cell-intrinsic responses against intracellular bacteria, including *Mycobacterium tuberculosis* (*Mayer-Barber et al., 2010*; *2011*). Additionally, immunosuppressive effects of type I IFNs may underlie their pharmacological efficacy in the treatment of multiple sclerosis (*Prinz et al., 2008*).

Given the contrasting immunosuppressive and antiviral functions of type I IFNs, we sought to directly test whether disabling AXL RTK signaling indeed translates into increased resistance to viral infection in vivo. Unexpectedly, $Axl^{-/-}$ mice were more susceptible than WT mice to influenza A virus (IAV) infection. This enhanced susceptibility correlated with reduced maturation of DCs and deficient induction of antiviral T cell responses. A similar impairment in inducing an effective adaptive T cell response in $Axl^{-/-}$ mice was detected during infection with the unrelated neurotropic West Nile virus (WNV). The failure to engage antiviral adaptive immunity could be ascribed to increased type I IFN and the associated reduction in IL-1β production in infected $Axl^{-/-}$ mice. Neutralization of type I IFN function restored the production of IL-1β in infected $Axl^{-/-}$ DCs and rescued the capacity of $Axl^{-/-}$ mice to induce the protective antiviral adaptive immune response and resist IAV infection. Similarly, delivery of IL-1β restored antiviral adaptive immunity in $Axl^{-/-}$ mice and survival to IAV infection. In summary, our studies underscore the function of AXL in calibrating the antiviral versus the immunosuppressive functions of type I IFNs during viral infection.

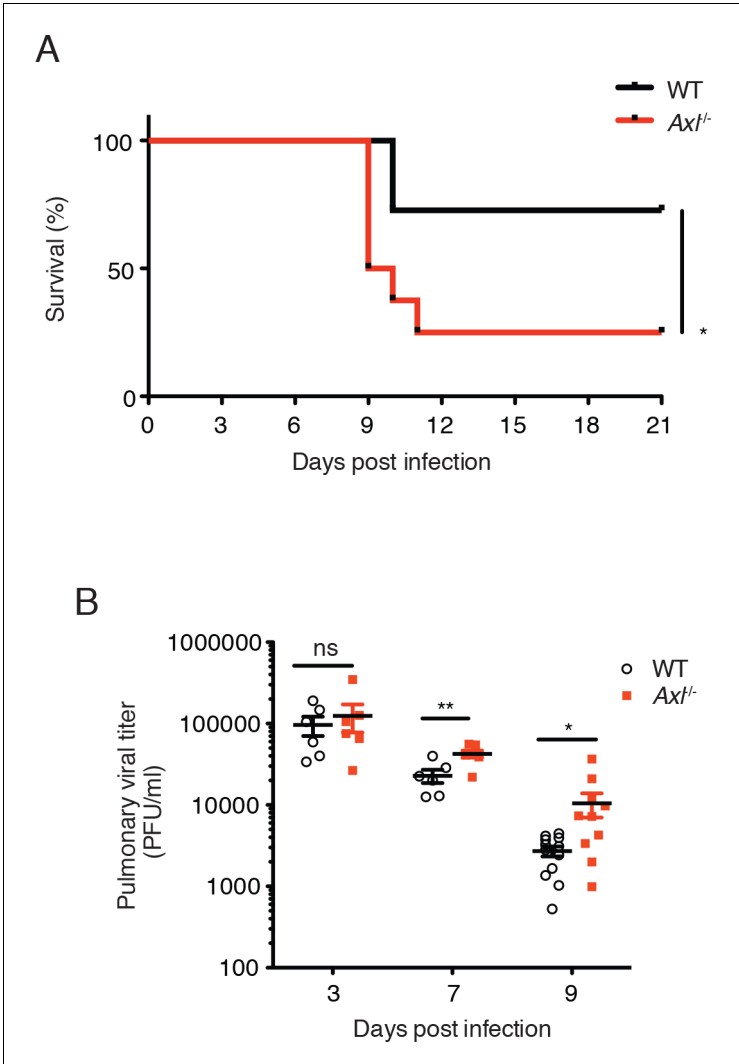

**Figure 1.** Loss of *Axl* increases susceptibility to influenza A virus infection in vivo. (**A**) Kaplan-Meier survival curves for wild-type (WT) and *Axl*[-/-] mice infected with 10 PFU of A/PR8 virus, 8–11 mice of each genotype and representative of 5 independent experiments. (**B**) Viral titers in the bronchoalveolar lavage (BAL) of WT and *Axl*[-/-] mice on days 3, 7, and 9 post infection with 10 PFU of PR8, as determined by qPCR of PR8 polymerase acidic protein (*PA*) RNA. PFU = plaque forming units. 6–12 mice were used per condition. ns, non-significant; *p<0.05; **p<0.01.

## Results

### Genetic ablation of *Axl* results in increased resistance to infection in DCs but overall enhanced susceptibility to IAV infection

To better understand the function of AXL during the course of IAV infection in vivo, mice were challenged with 10 PFU of A/Puerto Rico/8/1934 (H1N1) (PR8) and monitored for clinical signs of disease. By 11 days after intranasal administration of PR8, significantly more *Axl*[-/-] mice than WT mice succumbed to the infection (*Figure 1A*). This result is in agreement with a recent report by Fujimori et al (*Fujimori et al., 2015*). The increased susceptibility of *Axl*[-/-] mice to IAV infection correlated with higher viral titers in the bronchoalveolar lavage (BAL) fluid than in WT mice 7 and 9 days post-infection (*Figure 1B*), corresponding to when the CD8[+] T cell response is critical in viral clearance. However, no significant differences in viral loads were detected during the early phase of the infection between WT and *Axl*[-/-] mice (day 3 post-infection, *Figure 1B*).

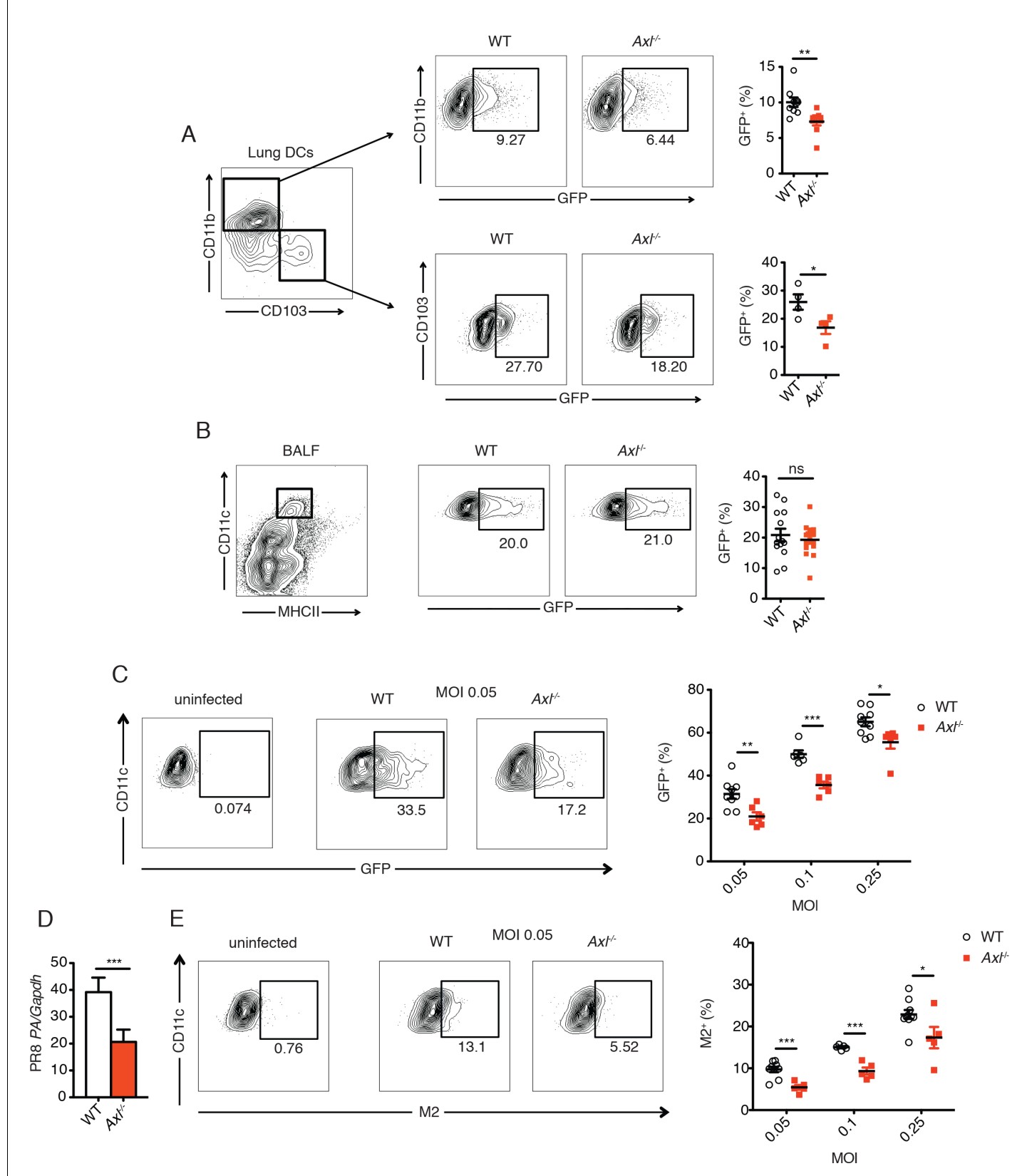

**Figure 2.** Genetic ablation of *Axl* confers resistance to IAV infection in dendritic cells in vivo and in vitro. WT and *Axl⁻/⁻* mice were infected with 3x10⁶ PFU of PR8-GFP for 72 hr and lung DCs were identified by flow cytometry. (**A**) Top, representative flow cytometry plots (left) and percentage of

*Figure 2 continued on next page*

*Figure 2 continued*

GFP$^+$CD11c$^+$MHCII$^+$CD11b$^+$ DCs (right) in infected WT and *Axl*$^{-/-}$ mice. n = 9 for each genotype, representing 3 independent experiments. Bottom, representative plots (left) and percentage of GFP$^+$CD11c$^+$MHCII$^+$CD103$^+$ DCs (right) in infected WT and *Axl*$^{-/-}$ mice. n = 4 for each genotype, representative of 3 independent experiments. (B) Representative flow cytometry plots (left) and percentage of GFP$^+$ alveolar macrophages (right) in infected WT and *Axl*$^{-/-}$ mice. 14–16 mice per genotype, 3 independent experiments. (C) WT and *Axl*$^{-/-}$ BMDCs were infected with PR8-GFP with indicated multiplicities of infection (MOIs) for 12 hr. Representative flow cytometry plots (left) and percentage of GFP$^+$ BMDCs (right) are shown. (D) Abundance of PR8 *PA* RNA normalized to *Gapdh* in WT and *Axl*$^{-/-}$ BMDCs after 12 hr of infection with 0.25 MOI of PR8-GFP, as determined by qPCR. (E) WT and *Axl*$^{-/-}$ BMDCs were infected as in (C). Representative plots (left) and percentage of IAV M2 ion channel$^+$ BMDCs (right) are shown. For (C) and (E), 5–9 samples were tested in each condition. Data are shown as representative or as the mean ± SEM of at least 4 independent samples per group representative of 4 independent experiments. ns, non-significant; *p<0.05; **p<0.01; ***p<0.001.

The following figure supplements are available for figure 2:

**Figure supplement 1.** AXL and MERTK expression in naive lung dendritic cells and alveolar macrophages.

**Figure supplement 2.** Total number of CD11c$^+$MHCII$^+$CD11b$^+$CD103$^-$ and CD11c$^+$MHCII$^+$CD11b$^-$CD103$^+$ cells in the lung 72 hr post infection with 3x10$^6$ PFU A/PR8 NS1-GFP.

**Figure supplement 3.** *Axl*$^{-/-}$ mice have fewer IAV-infected lung DCs than WT mice.

In contrast to these in vivo observations, previous studies have reported increased resistance to infection by other viruses in AXL-deficient DCs in vitro (*Bhattacharyya et al., 2013*; *Meertens et al., 2012*; *Morizono et al., 2011*; *Shimojima et al., 2007*; *2012*). Therefore, we tested whether *Axl*$^{-/-}$ DCs were more or less susceptible to IAV in vivo by using a recombinant strain of PR8 carrying a GFP reporter gene in the NS segment (PR8-GFP) (*Manicassamy et al., 2010*) and analyzing percentages of GFP$^+$ lung DCs 3 days post-infection. Two subsets of pulmonary DCs, CD11c$^+$CD11b$^+$-CD103$^-$ and CD11c$^+$CD11b$^-$CD103$^+$, have been identified as responsible for presenting and cross-presenting IAV antigens (*Ballesteros-Tato et al., 2010*; *Helft et al., 2012*; *Kim et al., 2014*). Flow cytometry analyses revealed that AXL is expressed in both of these DC subsets during influenza infection (*Figure 2—figure supplement 1*). Importantly, AXL ablation did not affect the total number of CD11c$^+$CD11b$^+$CD103$^-$ and CD11c$^+$CD11b$^-$CD103$^+$ DCs in the lung (*Figure 2—figure supplement 2*). When PR8-GFP infected *Axl*$^{-/-}$ mice were compared to WT mice, we detected significantly fewer infected CD11c$^+$CD11b$^+$CD103$^-$ DCs (*Figure 2A*, *Figure 2—figure supplement 3*). Likewise, fewer GFP$^+$ CD11c$^+$CD11b$^-$CD103$^+$ DCs were identified in *Axl*$^{-/-}$ mice (*Figure 2A*, *Figure 2—figure supplement 3*).

Another important cell type in the anti-IAV response is the alveolar macrophage (*Iwasaki and Pillai, 2014*). *Axl*$^{-/-}$ and WT alveolar macrophages were equally susceptible to infection by PR8-GFP (*Figure 2B*). Alveolar macrophages express both AXL and the related receptor MERTK, while only AXL but not MERTK was detected on lung DCs (*Fujimori et al., 2015*) and *Figure 2—figure supplement 1*). Thus, it is possible that MERTK compensates for the loss of AXL in alveolar macrophages, that MERTK is the relevant TAM receptor or that neither AXL nor MERTK regulate the susceptibility of alveolar macrophages to PR8-GFP.

Bone marrow-derived (BM)-DCs also express AXL (*Rothlin et al., 2007*). These cells were infected in vitro with a range of multiplicities of infection (MOI) of PR8-GFP and the degree of infection was measured as the percentage of GFP$^+$ BMDCs. Similar to a previous report using pseudotyped HIV-1 and WNV (*Bhattacharyya et al., 2013*), *Axl*$^{-/-}$ BMDCs were significantly more resistant to infection by PR8-GFP than WT BMDCs (*Figure 2C*). Additionally, the abundance of transcript of PR8 polymerase acidic protein (*PA*) in *Axl*$^{-/-}$ BMDCs was lower than in WT BMDC cultures (*Figure 2D*).

AXL is a well-established phagocytic receptor that mediates the engulfment of apoptotic cells (*Rothlin et al., 2015*; *Zagorska et al., 2014*). The AXL agonist, GAS6, can bind to phosphatidylserine exposed on the surface of apoptotic membranes and thus bridge apoptotic cells to AXL-expressing phagocytes. To rule out the possibility that the difference in GFP signal between WT and *Axl*$^{-/-}$ BMDCs was due to uptake of infected GFP$^+$ apoptotic cells, we measured the expression of IAV M2 ion channel on the cell surface. Newly synthesized IAV M2 channel is transported to the plasma membrane of infected cells for incorporation into the envelope of budding virions. Therefore, membrane-associated M2 is a marker of active infection. We detected reduced percentages of M2$^+$*Axl*$^{-/-}$

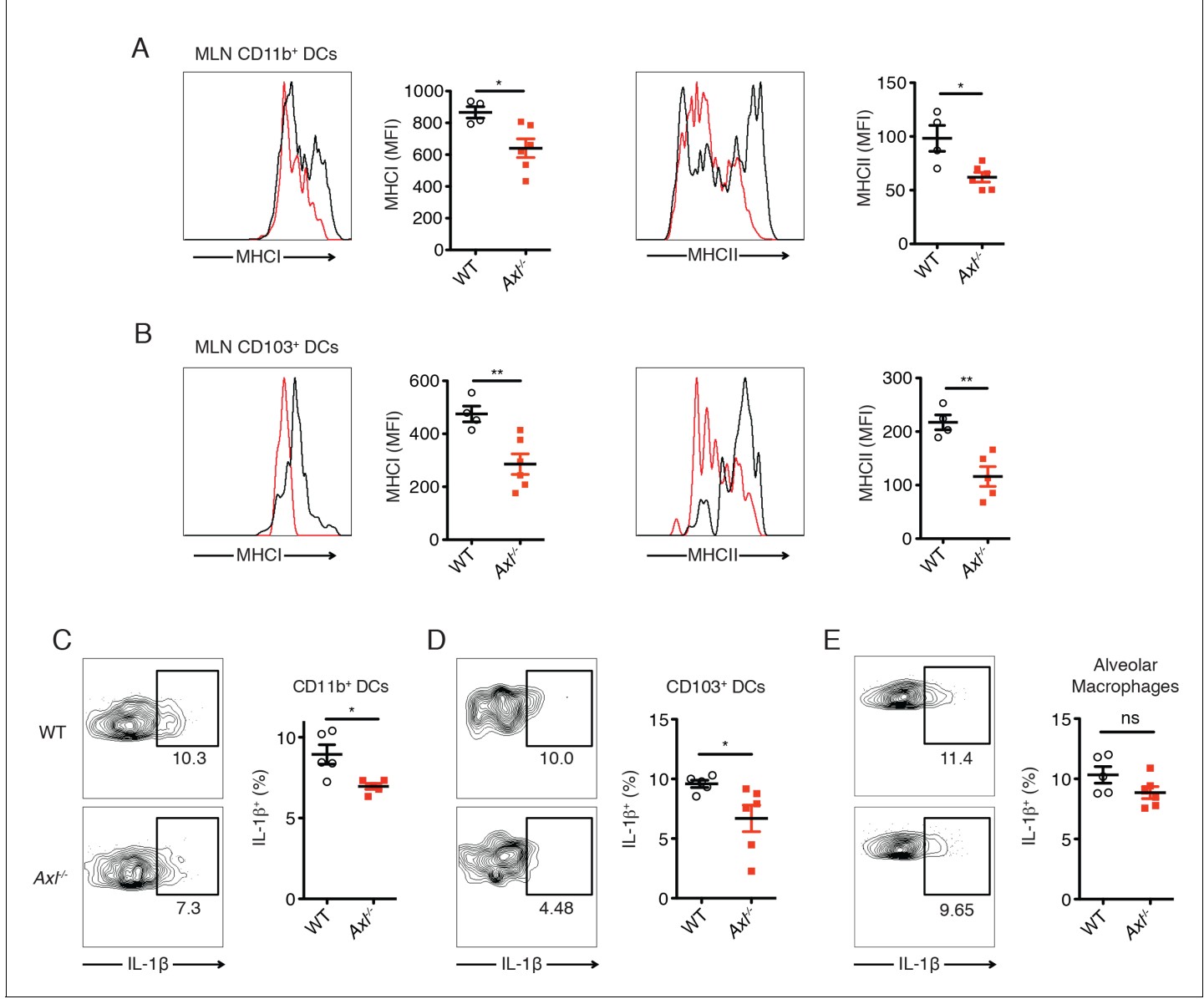

**Figure 3.** DCs in *Axl*$^{-/-}$ mice are less activated and produce less IL-1β than WT mice during IAV infection. (**A**) Expression of MHC-I and MHC-II molecules on CD11c$^+$MHCII$^+$CD11b$^+$CD103$^-$ mediastinal lymph node (MLN) DCs after 72 hr of infection with 3x10$^6$ PFU of PR8-GFP IAV as detected by flow cytometry. (**B**) Expression of MHC-I and MHC-II molecules on CD11c$^+$MHCII$^+$CD11b$^-$CD103$^+$ MLN DCs in mice infected as in (**A**). (**C**) Intracellular staining of IL-1β in lung CD11b$^+$ DCs 72 hr post infection with 3x10$^6$ PFU of PR8-GFP. (**D**) Intracellular staining of IL-1β in lung CD103$^+$ DCs infected as in (**C**). Data are presented as the mean ± SEM of 4–6 mice per condition, representative of 2–4 independent experiments. ns, non-significant; *p<0.05; **p<0.01.

BMDCs in comparison to M2$^+$ WT BMDCs throughout the range of tested MOIs of PR8-GFP (*Figure 2E*). Collectively, our results recapitulate the previously described resistance of *Axl*$^{-/-}$DCs to viral infection, but do not translate into improved antiviral response during infection in vivo.

## DC maturation and IL-1β production are impaired in *Axl*$^{-/-}$ mice during IAV infection

The induction of protective antiviral CD4$^+$ and CD8$^+$ T cell responses to IAV requires antigen presentation by DCs on MHC-II and MHC-I, respectively. In agreement with the increased resistance of lung DC subsets to IAV infection in *Axl*$^{-/-}$ mice, we detected a reduced maturation of these cells in

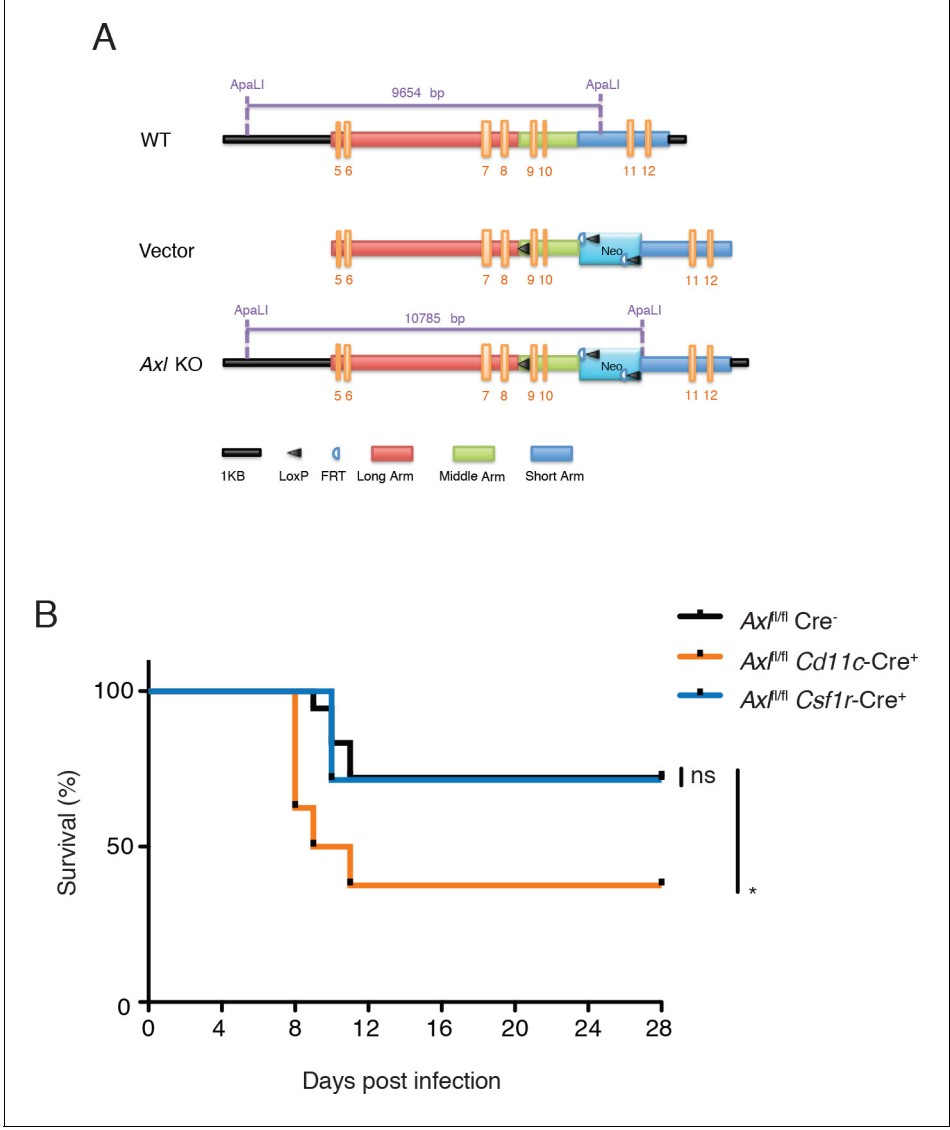

**Figure 4.** *Cd11c*-Cre⁺*Axl*^fl/fl mice but not *Csf1r*-Cre⁺*Axl*^fl/fl mice succumb to IAV infection. (**A**) Cloning strategy for the generation of Axl-floxed mice. *Axl*^fl/fl mice were subsequently crossed with *Cd11c*-Cre or *Csf1r*-Cre mice. (**B**) Kaplan-Meier survival curves for Cre⁻*Axl*^fl/fl, *Cd11c*-Cre⁺*Axl*^fl/fl, and *Csf1r*-Cre⁺*Axl*^fl/fl mice infected with 10 PFU of A/PR8 virus, 7–18 mice per group and representative of 2 independent experiments. ns, non-significant; *p<0.05.

The following figure supplements are available for figure 4:

**Figure supplement 1.** AXL expression by immune cells in the lung during IAV infection.

**Figure supplement 2.** AXL is selectively ablated in *Cd11c*-Cre⁺*Axl*^fl/fl and *Csf1r*-Cre⁺*Axl*^fl/fl mice.

**Figure supplement 3.** *Cd11c*-Cre⁺*Axl*^fl/fl BMDCs are resistant to IAV infection.

**Figure supplement 4.** Cre⁻*Axl*^fl/fl and *Cd11c*-Cre⁺*Axl*^wt/wt mice clear IAV infection.

the mediastinal lymph nodes (MLNs). Significantly lower amounts of MHC-I and MHC-II were measured on CD11c⁺CD11b⁺CD103⁻ DCs in the draining MLN in *Axl*⁻/⁻ mice 72 hr post-infection with IAV (**Figure 3A**). The reduced expression of MHC-I and MHC-II was also observed in *Axl*⁻/⁻ CD11c⁺-CD11b⁻CD103⁺ cells (**Figure 3B**). IL-1β has been shown to be required for effective activation of

lung dendritic cells and induction of adaptive immunity during IAV infection (*Pang et al., 2013*). We found significantly fewer IL-1β-producing CD11c⁺CD11b⁺CD103⁻ and CD11c⁺CD11b⁻CD103⁺ DCs in the lung of *Axl⁻/⁻* mice 72 hr post-infection in comparison to WT mice (*Figure 3C and D*). In contrast, alveolar macrophages from both infected WT and *Axl⁻/⁻* mice produced equal amounts of IL-1β (*Figure 3E*).

## Myeloid cell-specific ablation of *Axl* is sufficient to render mice more susceptible to IAV infection

AXL expression is not limited to DCs and macrophages—it is also detected on mature NK cells during viral infection (*Figure 4—figure supplement 1*) and non-hematopoietic cells (*Rothlin et al., 2015*). To test whether the loss of AXL expression on myeloid cells was sufficient to lead to increased susceptibility to IAV infection, we generated *Axl*^fl/fl^ mice (*Figure 4A*) and crossed them to *Cd11c*-Cre (*Caton et al., 2007*). CD11c is a classic marker of DCs (*Caton et al., 2007*) and it is also expressed by alveolar macrophages. Ablation of AXL was confirmed in lung DCs and alveolar macrophages in *Cd11c*-Cre⁺ *Axl*^fl/fl^ mice, while its expression remained intact in NK cells (*Figure 4—figure*

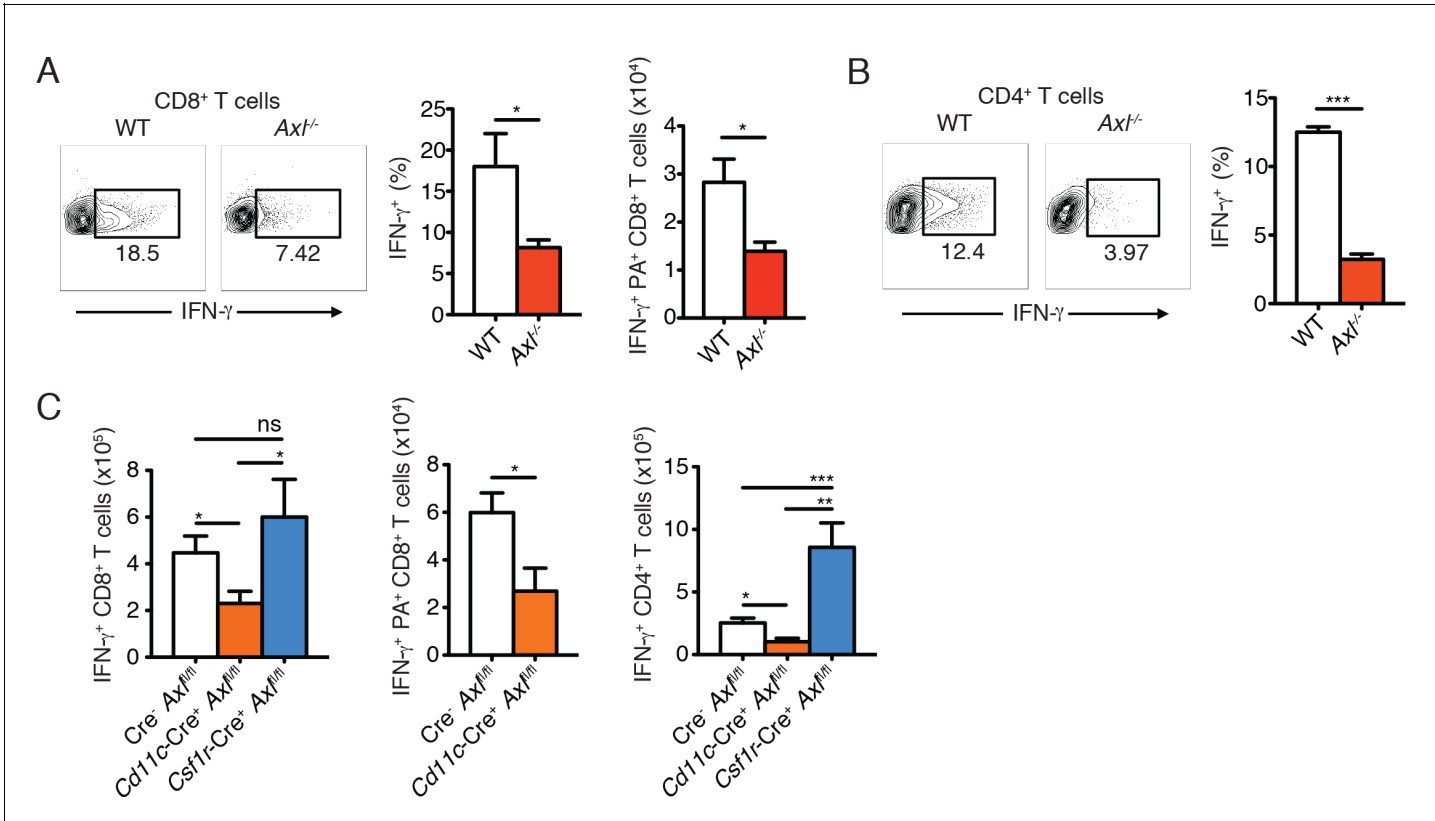

**Figure 5.** *Axl⁻/⁻* mice and *Cd11c*-Cre⁺*Axl*^fl/fl^ mice mount impaired T cell responses to IAV infection. (**A**) Representative plots (left) and percentage (middle) of CD8⁺IFN-γ⁺ T cells in the lung of WT and *Axl⁻/⁻* mice after 9 days of infection with 10 PFU of PR8. 4–5 mice per genotype, representative of 4 independent experiments. Right, quantification of IFN-γ-producing H-2D^b^-restricted CD8⁺ T cells specific for IAV PA amino acids 224–233 in the lung 9 days post infection with 10 PFU of PR8. 7–8 mice per genotype, 2 independent experiments. (**B**) Representative plots (left) and percentage (right) of CD4⁺IFN-γ⁺ T cells in the lung of WT and *Axl⁻/⁻* mice infected as in (**A**). (**C**) Number of CD8⁺IFN-γ⁺ T cells (left), CD8⁺PA⁺IFN-γ⁺ T cells (middle), and CD4⁺IFN-γ⁺ (right) in the lung 9 days post-infection with 10 PFU of PR8 in Cre⁻*Axl*^fl/fl^, *Cd11c*-Cre⁺*Axl*^fl/fl^, and *Csf1r*-Cre⁺*Axl*^fl/fl^ mice, as indicated. 5–10 mice per genotype, representative of 2–3 independent experiments. ns, non-significant; *p<0.05; **p<0.01; ***p<0.001

The following figure supplements are available for figure 5:

**Figure supplement 1.** *Axl⁻/⁻* mice display an early defect in CD8⁺ T cell activation during PR8 infection.

**Figure supplement 2.** T cells of *Axl⁻/⁻* and *Cd11c*-Cre⁺*Axl*^fl/fl^ mice have reduced CD44 expression during IAV infection.

supplement 2). BMDCs derived from *Cd11c*-Cre$^+$ *Axl*$^{fl/fl}$ recapitulated the increased resistance to PR8-GFP infection characteristic of *Axl*$^{-/-}$ BMDCs, confirming the functional ablation of AXL in this line (*Figure 4—figure supplement 3*). *Axl*$^{fl/fl}$ mice were also crossed to *Csf1r*-Cre mice (*Deng et al., 2010*). CSF1 receptor is preferentially expressed in macrophages, although it can also be detected in CD11b$^+$ but not CD103$^+$ lung DCs (*Ginhoux et al., 2009*). In agreement with this reported expression pattern, *Csf1r*-Cre$^+$ *Axl*$^{fl/fl}$ featured preferential ablation of AXL in alveolar macrophages than lung DCs (*Figure 4—figure supplement 2*).

Next, we challenged *Cd11c*-Cre$^+$*Axl*$^{fl/fl}$, *Csf1r*-Cre$^+$*Axl*$^{fl/fl}$ and respective controls to IAV infection. Analogous to that seen in *Axl*$^{-/-}$ mice, *Cd11c*-Cre$^+$*Axl*$^{fl/fl}$ mice succumbed at a higher frequency to PR8 than *Cd11c*-Cre$^-$*Axl*$^{fl/fl}$ control mice (*Figure 4B*). The sensitivity of *Cd11c*-Cre$^+$*Axl*$^{fl/fl}$ to PR8 infection was not due to Cre expression, as *Cd11c*-Cre$^+$*Axl*$^{wt/wt}$ mice were not more susceptible to infection than WT mice (*Figure 4—figure supplement 4*). In contrast, *Csf1r*-Cre$^+$ *Axl*$^{fl/fl}$ were as resistant to IAV infection as control mice (*Figure 4B*). Take together, these results indicate that the ablation of AXL in myeloid cells is sufficient to confer susceptibility to IAV infection and that preserved expression in DCs appears to be required to resist the infection.

## Ablation of AXL in myeloid cells impairs induction of antiviral adaptive immunity

Clearance of IAV depends on the optimal activation of the adaptive immune response (*Iwasaki and Pillai, 2014*; *Strutt et al., 2013*; *Sun and Braciale, 2013*). We therefore investigated the induction of adaptive antiviral immunity in the absence of AXL. Expression of CD69, an early T cell activation marker, was significantly lower in draining MLN CD8$^+$ T cells in *Axl*$^{-/-}$ mice in comparison to WT controls 3 days post-infection with PR8 (*Figure 5—figure supplement 1*). Lung CD8$^+$ T cells also showed a diminished production of IFN-γ 9 days post infection (*Figure 5A*). The number of IFN-γ$^+$ antigen-restricted CD8$^+$ T cells specific for IAV PA amino acids 224–233 was similarly reduced in the lung of *Axl*$^{-/-}$ mice (*Figure 5A*). Furthermore, as an additional marker of activation, the number of CD8$^+$CD44$^+$ cells in the MLN as well as the expression level of CD44 on CD8$^+$ T cells in the lung was less in *Axl*$^{-/-}$ mice compared to WT mice (*Figure 5—figure supplement 2*). Similarly, CD4$^+$ T cells in *Axl*$^{-/-}$ mice were less activated as evidenced by their reduced expression of IFN-γ and CD44 (*Figure 5B* and *Figure 5—figure supplement 2*).

As was seen in *Axl*$^{-/-}$ mice, *Cd11c*-Cre$^+$*Axl*$^{fl/fl}$ but not *Cd11c*-Cre$^-$*Axl*$^{fl/fl}$ had significantly fewer number of IFN-γ producing CD8$^+$ T cells (*Figure 5C*). This reduction was in part due to lower numbers of PA-specific CD8$^+$ T cells (*Figure 5C*). In contrast, *Csf1r*-Cre$^+$*Axl*$^{fl/fl}$ mice had preserved CD8$^+$ T cell responses. (*Figure 5C*). Similarly, *Cd11c*-Cre$^+$*Axl*$^{fl/fl}$ mice developed fewer IFN-γ producing CD4$^+$ T cells in comparison to respective controls, while this response was conserved in *Csf1r*-Cre$^+$-*Axl*$^{fl/fl}$ mice (*Figure 5C*). The number of CD8$^+$CD44$^+$ and CD4$^+$CD44$^+$ T cells was also significantly reduced in the lung of *Cd11c*-Cre$^+$*Axl*$^{fl/fl}$ infected mice compared to control mice (*Figure 5—figure supplement 2*). These results indicate that myeloid-specific ablation of AXL in *Cd11c*-Cre$^+$*Axl*$^{fl/fl}$ mice is sufficient to account for the impaired T cell activation phenotype seen in complete *Axl*$^{-/-}$ mice. Furthermore, the ability of *Csf1r*-Cre$^+$*Axl*$^{fl/fl}$ infected mice to mount protective adaptive antiviral responses is consistent with the preserved expression of AXL in lung DCs and the lack of increased susceptibility to infection in this conditional knock out line.

To corroborate the findings in the context of an unrelated virus, *Axl*$^{-/-}$ mice were infected subcutaneously with WNV and spleens were collected 8 days after infection. As seen during IAV infection, *Axl*$^{-/-}$ mice developed deficient CD8$^+$ T cell responses to WNV. The percentage and number of NS4B tetramer$^+$ cells (*Figure 6A*) and IFN-γ-producing CD8$^+$ T cells (*Figure 6B*) were reduced in *Axl*$^{-/-}$ mice in comparison to WT mice. Similarly, levels of intracellular granzyme B were reduced in both NS4B tetramer$^+$*Axl*$^{-/-}$ CD8$^+$ T cells and total *Axl*$^{-/-}$ CD8$^+$ T cell populations (*Figure 6C*). These results are in agreement with the increased susceptibility of *Axl*$^{-/-}$ mice to WNV infection (*Miner et al., 2015*). Collectively, these results show that loss of AXL signaling leads to a defect in priming the adaptive antiviral T cell responses after IAV and WNV infections.

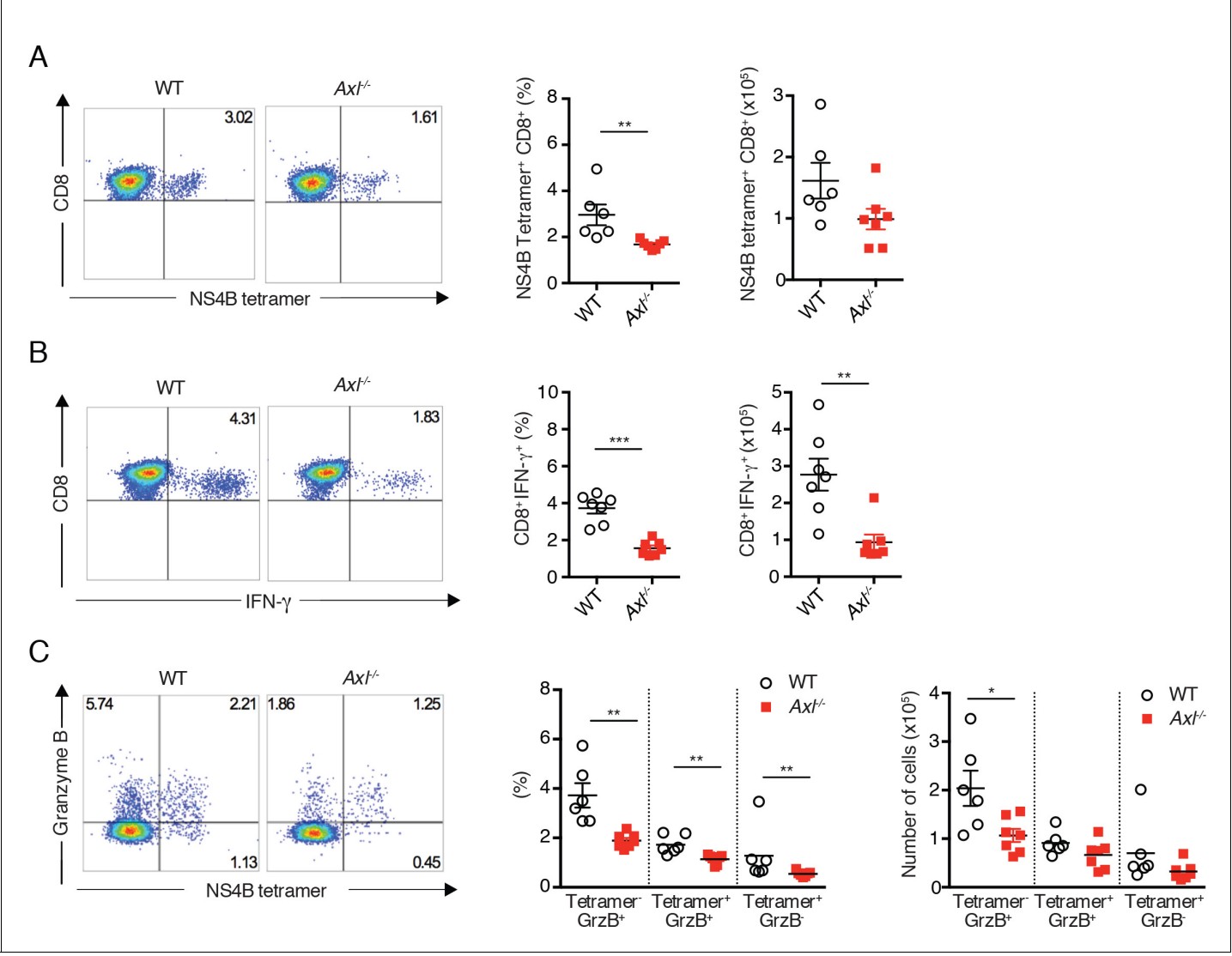

**Figure 6.** *Axl⁻/⁻* mice mount a deficient CD8⁺ T cell response to WNV infection. WT and *Axl⁻/⁻* mice were infected subcutaneously with $10^2$ PFU of WNV, and spleens were harvested 8 days post infection after extensive cardiac perfusion with PBS. (**A**) Representative flow cytometry plots (left) and percentage and number (right) of NS4B tetramer⁺ CD8⁺ T cells. (**B**) Representative flow cytometry plots (left) and percentage and number (right) of CD8⁺IFN-γ⁺ T cells. (**C**) Representative flow cytometry plots (left) and percentage and number (right) of CD8⁺ T cells stained for NS4B tetramer and granzyme B. Data are presented as the mean ± SEM of 6–7 mice per genotype. Data are pooled from two independent experiments. p<0.05; **p<0.01; ***p<0.001.

## Neutralization of type I IFN signaling or administration of recombinant IL-1β restores antiviral immunity in *Axl⁻/⁻* mice

AXL is a negative regulator of type I IFN production and genetic ablation of *Axl* has been shown to lead to increased production of type I IFNs upon viral infection (*Bhattacharyya et al., 2013*; *Rothlin et al., 2007*). We detected increased production of IFN-β in IAV-infected *Axl⁻/⁻* versus WT BMDC cultures (*Figure 7A*). Furthermore, neutralization of type I IFN signaling by MAR1-5A3 anti-IFNAR antibody restored the susceptibility of *Axl⁻/⁻* BMDCs to PR8 infection (*Figure 7B*). Next, we tested whether the enhanced type I IFN response in *Axl⁻/⁻* mice accounted for their increased susceptibility to IAV infection. *Axl⁻/⁻* and WT mice were injected with MAR1-5A3 or the respective isotype control one day prior to infection with PR8. We detected a significant increase in the survival of IAV-infected *Axl⁻/⁻* mice treated with MAR1-5A3 (*Figure 7C*). This correlated with a restoration of IFN-γ⁺ PA-restricted CD8⁺ T cells and IFN-γ producing CD4⁺ T cells (*Figure 7D*). Similarly, the number of

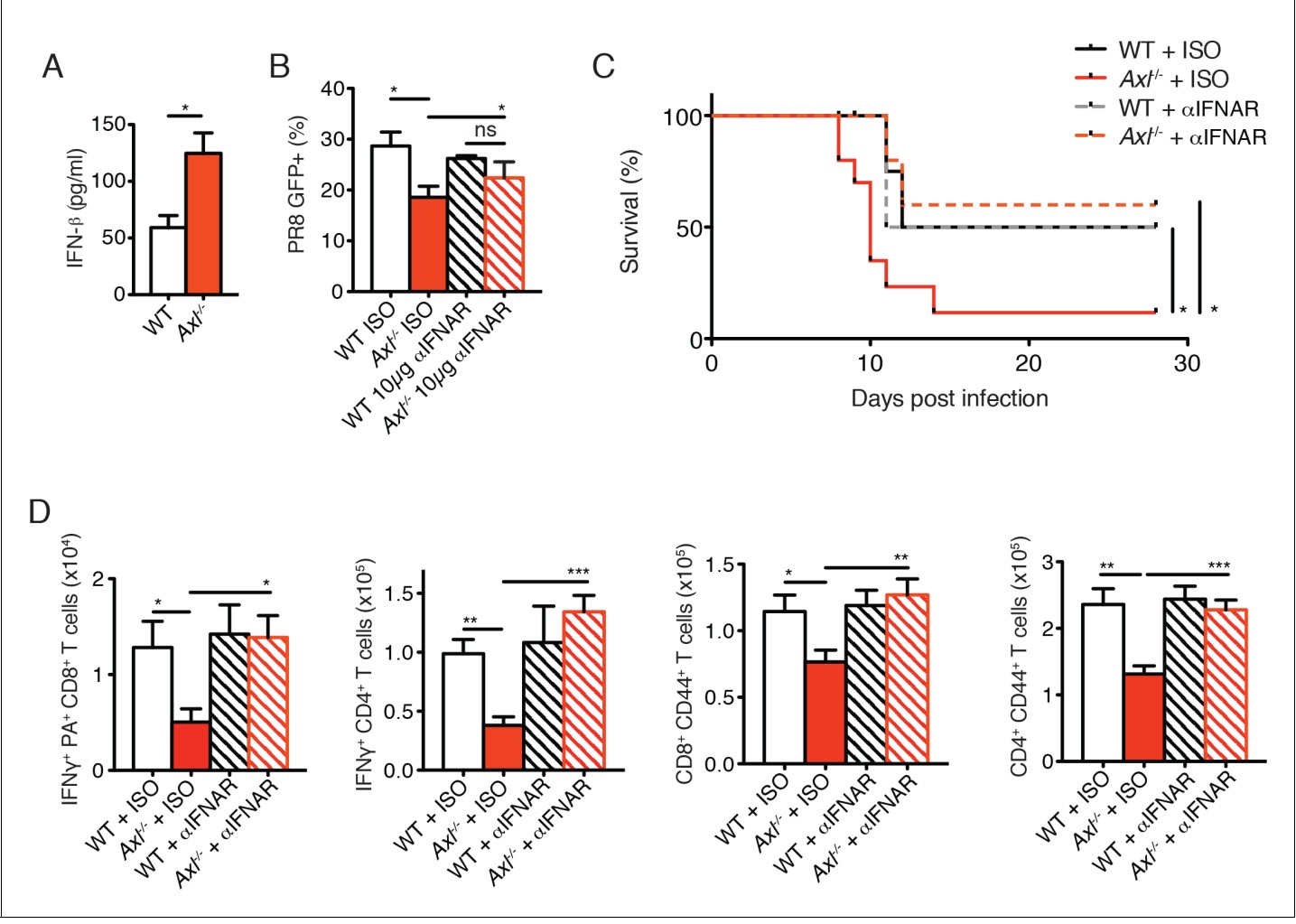

**Figure 7.** Blockade of IFNAR signaling protects *Axl*[-/-] mice to IAV infection and rescues T cell activation. (**A**) IFN-β in the supernatant of WT and *Axl*[-/-] BMDCs after 12 hr of infection with 0.25 MOI of PR8, as determined by ELISA from 4 independent experiments. (**B**) Percentage of GFP+ WT and *Axl*[-/-] BMDCs infected with 0.05 MOI PR8 for 12 hr treated with 10 μg/ml of IgG$_1$ isotype control or α-IFNAR MAR1-5A3 antibody. Data is compiled from 3 independent experiments. (**C**) Kaplan-Meier survival curves for WT and *Axl*[-/-] mice given α-IFNAR MAR1-5A3 antibody or isotype control by IP injection one day prior to infection with 10 PFU of A/PR8 virus, 8–10 mice per group, 2 independent experiments. (**D**) WT and *Axl*[-/-] mice were treated with antibody and infected as in (**C**). Number of IFN-γ-producing PA tetramer+ CD8+ T cells (left) and IFN-γ+ CD4+ T cells (middle) in the lung 7 days post infection with 10 PFU of PR8. 4–5 mice in each group, representative of 2 independent experiments. Number of CD8+CD44+ T cells (middle) and CD4+CD44+ T cells (right) in the MLN 9 days post-infection with 10 PFU of PR8. 8–10 mice per group, representing 2 independent experiments. Data are shown as the mean ± SEM. *p<0.05; **p<0.01; ***p<0.001.

CD8+CD44+ and CD4+CD44+ T cells in infected *Axl*[-/-] mice treated with the anti-type I IFN receptor antibody were restored to that of control mice (**Figure 7D**).

The immunosuppressive properties of type I IFNs are mediated, in part, by their ability to block the production of IL-1β (*Guarda et al., 2011*; *Mayer-Barber et al., 2010*; *2011*). This is particularly relevant in the context of IAV infection, as IL-1β is required for effective priming of antiviral T cells and antibody responses (*Ichinohe et al., 2009*; *Pang et al., 2013*; *Schmitz et al., 2005*). The increased production of IFN-β in IAV-infected *Axl*[-/-] BMDCs correlated with a concomitant reduction in the production of IL-1β (*Figure 8A*). This is in agreement with the decreased production of IL-1β found in lung DCs of infected *Axl*[-/-] mice (*Figure 3C and D*). Furthermore, neutralization of type I IFN signaling in infected *Axl*[-/-] BMDCs rescued IL-1β production by these cells (*Figure 8B*). To test if the diminished production of IL-1β in infected *Axl*[-/-] mice was causal for their increased susceptibility to IAV, we administered recombinant IL-1β intranasally to WT and *Axl*[-/-] mice on days 1, 2, and 3

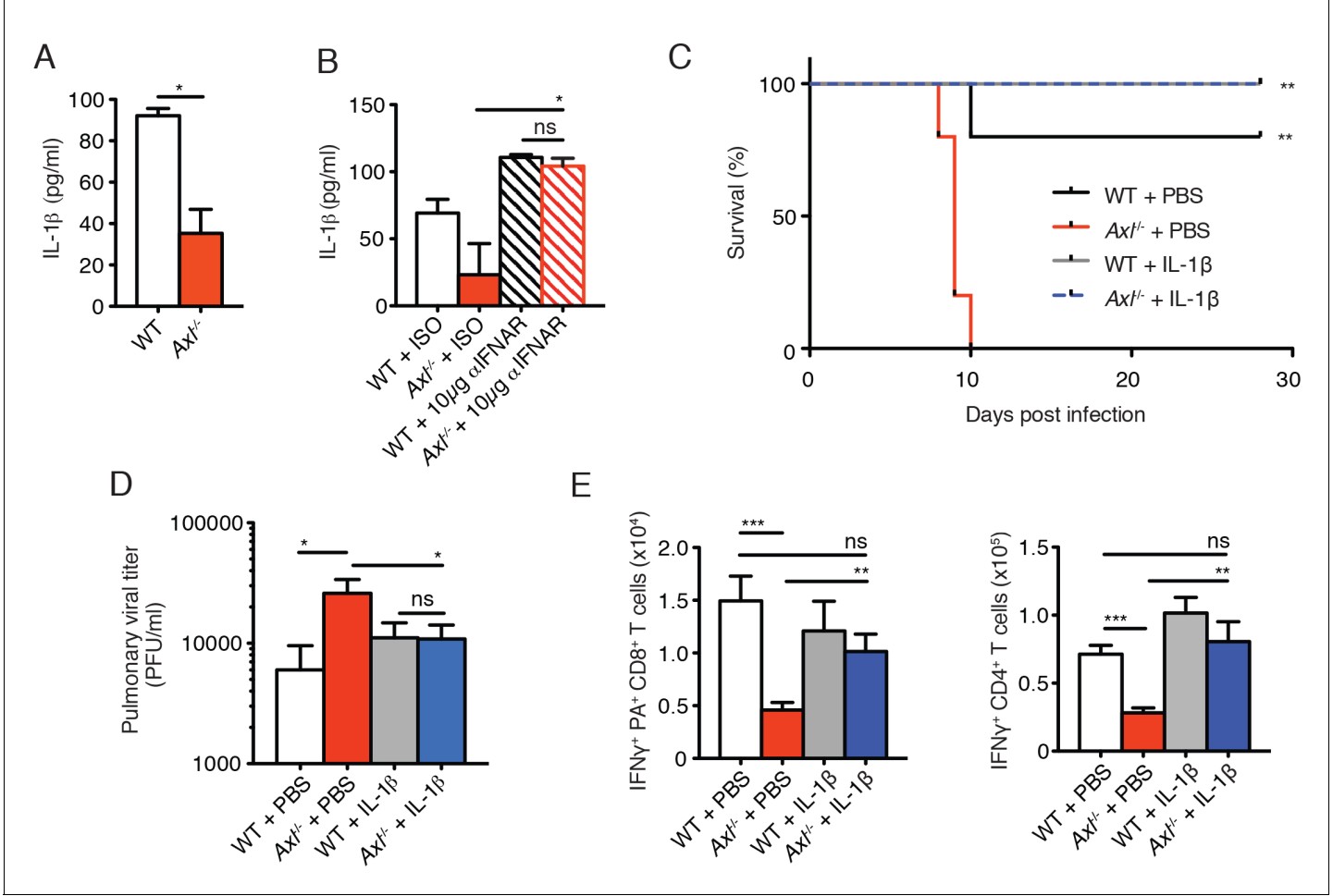

**Figure 8.** Intranasal administration of IL-1β rescues *Axl*<sup>-/-</sup> T cell activation and confers protection to IAV infection. (**A**) IL-1β levels in supernatant of WT and *Axl*<sup>-/-</sup> BMDCs after 12 hr of infection with 0.25 MOI of PR8, as determined by ELISA from 4 independent experiments. (**B**) IL-1β in supernatant of WT and *Axl*<sup>-/-</sup> BMDCs infected with 0.05 MOI of PR8-GFP for 12 hr treated with 10 μg/ml of isotype control or α-IFNAR MAR1-5A3 antibody, as determined by ELISA from 3 independent experiments. (**C-E**) WT and *Axl*<sup>-/-</sup> mice were intranasally administered PBS or 20 ng of recombinant IL-1β on days 0, 1, 2, and 3 post infection with 10 PFU of PR8. (**C**) Kaplan-Meier survival curves for mice treated as indicated with 5 mice per group, representative of 4 independent experiments. **\**Axl*<sup>-/-</sup> mice given PBS succumbed to infection significantly more than the other experimental groups. (**D**) Viral titers in the bronchoalveolar lavage (BAL) collected 9 days post infection determined by qPCR of PR8 *PA* RNA. 6–10 mice per group, representing 3 independent experiments. (**E**) Number of IFN-γ-producing PA tetramer<sup>+</sup> CD8<sup>+</sup> T cells and IFN-γ<sup>+</sup> CD4<sup>+</sup> T cells in the lung 7 days post infection with PR8. 4–5 mice in each group, representative of 2 independent experiments. Data are shown as the mean ± SEM. *p<0.05; **p<0.01.

The following figure supplement is available for figure 8:

**Figure supplement 1.** Intranasal IL-1β delivery rescues *Axl*<sup>-/-</sup> T cell CD44 expression during IAV infection.

post-infection with 10 PFU of PR8. *Axl*<sup>-/-</sup> mice that were given IL-1β were completely protected and survived the infection (*Figure 8C*). Consistent with this protection, viral titers in the BALF of *Axl*<sup>-/-</sup> mice given IL-1β were controlled to the level of WT controls (*Figure 8D*). Furthermore, the administration of IL-1β to *Axl*<sup>-/-</sup> mice restored the antiviral adaptive immune response as measured by the number of IFN-γ<sup>+</sup> PA-restricted CD8<sup>+</sup> T cells and IFN-γ producing CD4<sup>+</sup> T cells (*Figure 8E*) as well as CD8<sup>+</sup>CD44<sup>+</sup> and CD4<sup>+</sup>CD44<sup>+</sup> T cells (*Figure 8—figure supplement 1*). In summary, these results demonstrate that the genetic ablation of *Axl* leads to enhanced production of type I IFNs and decreased production of IL-1β resulting in impaired induction of antiviral adaptive immunity and clearance of virus.

## Discussion

In vitro experiments have led to the speculation that AXL promotes the infection of several enveloped viruses including vaccinia (*Morizono et al., 2011*), Lassa (*Shimojima et al., 2012*), dengue (*Meertens et al., 2012*), and WNV (*Bhattacharyya et al., 2013*). Since loss of AXL function in DCs protected the cells from viral infection, these results suggested that AXL inhibition might lead to improved antiviral response in infected hosts. This idea is of particular importance given that small molecule inhibitors against AXL are currently in development and one of them is in Phase 1 clinical trial (*Graham et al., 2014*). In direct opposition to this hypothesis, our experimental evidence demonstrates that mice featuring the genetic ablation of *Axl*, even after selective deletion in myeloid cells, are more susceptible to viral infection than WT mice.

We report that loss of *Axl* leads to a reduced ability to mount an adequate adaptive antiviral response, as exemplified by deficient priming of T cells after IAV and WNV infection. This phenotype is consistent with previous findings on the susceptibility to IAV infection of mice lacking CD8$^+$ T cells (*Bender et al., 1992*; *Wu et al., 2010*). How does AXL signaling protect the host against viral infection? Albeit paradoxical, the ability of AXL to inhibit type I IFN production appears to be important for induction of antiviral adaptive immunity. While type I IFNs are considered classical inducers of DC maturation (*Gallucci et al., 1999*; *Montoya et al., 2002*), the immunosuppressive effects of type I IFNs are also well known (*McNab et al., 2015*). Indeed, the increased production of type I IFNs in response to infection in *Axl$^{-/-}$* cells limited the production of IL-1β, a cytokine required for the effective priming of antiviral T cells (*Ichinohe et al., 2009*; *Pang et al., 2013*; *Schmitz et al., 2005*). Activation of the NLRP3 inflammasome and production of IL-1β are important components of the antiviral immune response to a variety of RNA viruses, including IAV, WNV, Sendai virus, adenovirus, and vaccinia virus (*Kanneganti, 2010*; *Ramos et al., 2012*). Similar to our findings in *Axl$^{-/-}$* mice, hosts deficient in inflammasome signaling experience heightened mortality to IAV or WNV infection (*Allen et al., 2009*; *Ichinohe et al., 2009*; *Ramos et al., 2012*; *Thomas et al., 2009*). Thus, our studies highlight the central role of AXL in the protection of the host against viral infections through the tightly regulated production of type I IFNs.

AXL is expressed not only in myeloid cells, but also in NK cells and non-hematopoietic cells (*Rothlin et al., 2015*). We generated an *Axl$^{fl/fl}$* mouse line and demonstrated that ablation of *Axl* in DCs and alveolar macrophages was sufficient to result in deficient T cell responses and confer susceptibility to IAV. While AXL function in DCs appears to be required for the appropriate priming of adaptive immunity, our approach did not ablate *Axl* in exclusively these cells. The development of more selective approaches to induce the ablation of genes in distinct DC populations will provide a better understanding. In contrast, mice featuring preferential ablation in macrophages did not succumb to the same infection. AXL and the related RTK MERTK are expressed in alveolar macrophages. These receptors are not only inhibitors of the immune response but also mediate the phagocytosis of apoptotic cells (*Fujimori et al., 2015*; *Rothlin et al., 2015*). Thus it is possible that AXL and MERTK function in alveolar macrophages is dispensable for the regulation of the priming of adaptive immunity, but participates in the clearance of apoptotic debris during the resolution phase.

Human myeloid cells, including DCs, express AXL (*Scutera et al., 2009*). Our study suggests that disabling this receptor function using small molecule inhibitors or blocking antibodies could lead to increased susceptibility to viral infections in humans rather than the desired increased resistance predicted by in vitro studies (*Bhattacharyya et al., 2013*; *Meertens et al., 2012*; *Morizono et al., 2011*; *Shimojima et al., 2007*). The prevalence of this AXL function in distinct viral infections should be carefully considered in the development of pharmacological tools that inhibit this RTK.

## Materials and methods

### Experimental procedures

#### Mice

Mice used in this study were age- and sex-matched in the C57BL/6 background. Mice were stratified according to sex and randomly allocated to different experimental groups. *Axl$^{-/-}$* mice were generated as previously described (*Lu et al., 1999*) and *Axl$^{fl/fl}$* mice were generated in C57BL/6 as described in *Figure 3A*. The neomycin cassette was removed during the electroporation of ES cells by using FLP C57/B6 ES cells. *Cd11c*-Cre mice (*Caton et al., 2007*) were obtained from Jackson

Laboratory. *Csf1r*-Cre mice were a gift from Jeffrey Pollard from the University of Edinburgh. Mice were housed in the Yale Animal Resource Center in specific-pathogen free facilities and treated according to IACUC (Institutional Animal Care and Use Committee) protocol or were approved and performed in accordance with the Institutional Animal Care and Use Committee at the Washington University School of Medicine (assurance number A3381-01).

## Bone marrow-derived dendritic cell preparation

Bone marrow-derived dendritic cells (BMDCs) were generated from bone marrow progenitor cells flushed from mouse femurs and tibias of gender- and age-matched donors. $2 \times 10^5$ progenitor cells/ ml were incubated at 37°C on 24-well cell culture plates in complete media containing RPMI 1640, 10% fetal bovine serum (FBS), 1% penicillin and streptomycin, and supplemented with granulocyte macrophage colony-stimulating factor (GM-CSF) at a concentration of 20 ng/ml (PeproTech, Rocky Hill, NJ). Fresh enriched media was added on days 3 and 6 of differentiation. BMDCs were subsequently used on day 7.

## Propagation of viral stocks and measuring of viral titers

A/PR8 (H1N1) and A/PR8 NS1-GFP (gift of Adolfo García-Sastre (*Manicassamy et al., 2010*)) were propagated for 2 days at 35°C in the allantoic cavities of 10- to 11-day old fertilized chicken eggs. BAL fluid from mice was collected for the measurement of viral titers at the indicated days post-infection by washing the trachea and lungs with 3 ml of PBS containing 0.1% BSA. Viral titers were quantified by standard viral plaque assay using Madin-Darby canine kidney (MDCK) cells or by qPCR quantification comparing samples to a standard curve generated from cDNA of RNA isolated from MDCK-titered stock. Viral stocks and BAL fluid samples were stored at -80°C. The WNV strain (3000.0259) was isolated in New York in 2000 and passaged once in C6/36 *Aedes albopictus* cells.

## In vitro infection of BMDCs

Prior to infection of BMDCs, media was aspirated and wells were washed once with PBS. Viral stock was diluted in 0.1% BSA in PBS to the indicated multiplicities of infection (MOIs). 100 μl of diluted virus was added to each well of BMDCs on the 24-well plates and placed in a 37°C incubator. Plates were lightly tapped every 20 min during a 1 hr incubation process to keep cells evenly covered by liquid. After 1 hr, virus was aspirated and wells were washed once with PBS. 1 ml of complete RPMI (RPMI 1640, 10% FBS, 1% penicillin and streptomycin) was added to each well and cultures were incubated at 37°C for 12 hr. Anti-mouse IFN-α/β receptor 1 (IFNAR1) antibody (MAR1-5A3, Leinco Technologies, Fenton, MO) or $IgG_1$ control was used at a concentration of 10 μg/ml and added to the complete media for the 12-hr incubation following infection in the indicated experiments. After infection, supernatants were stored at -80°C, and cells were collected by washing with ice-cold PBS.

## In vivo infections

Mice were anesthetized by intraperitoneal injection of ketamine and xylazine. 10 PFU PR8 or $3 \times 10^6$ PFU PR8-GFP was suspended in 20 μl and was intranasally administered. Weight change and overall appearance of health was monitored daily. Mice were sacrificed by $CO_2$ asphyxiation at the indicated time points or euthanized upon falling below 80% initial starting weight. For WNV studies, mice (8- to 10-week old, both sexes) were inoculated subcutaneously via footpad injection with $10^2$ PFU of WNV. In experiments where mice were treated with recombinant IL-1β, 20 ng of IL-1β (eBioscience, San Diego, CA) or PBS vehicle control were administered intranasally in 20 μl immediately following inoculation with virus. The same dose of IL-1β or PBS control was subsequently given days 1, 2, and 3 post-infection while mice were anesthetized by isolfurane. Mice treated with α-IFNAR MAR1-5A3 (Leinco Technologies) or $IgG_1$ isotype control (600 μg/mouse) were injected intraperitoneally one day prior to infection with PR8.

## Cell preparation and flow cytometry

To stain for flow cytometry, collected cells were washed once with PBS and then incubated with α-CD16/32 Fc-block clone 93 (Biolegend, San Diego, CA) in PBS for 10 min at room temperature. After washing again with PBS, cells were incubated with their respective antibody cocktail for 30 min at 4°C. Following subsequent washes, prepared cells were fixed with 1% paraformaldehyde. Anti-

mouse antibodies used in the study were purchased from BioLegend , BD-Biosciences, eBioscience, Santa Cruz Biotechnology (Dallas, TX), Invitrogen (Carlsbad, CA), Becton Dickinson (Franklin Lakes, NJ), or R&D Systems (Minneapolis, MN). Conjugated antibodies to FITC, PE, PE-Cy7, PerCP, PerCP-Cy5.5, APC, APC-Cy7, or Pacific Blue were used for flow cytometry to target CD45 (30-f11), CD11c (N418), CD11b (M1/70), H-2kb MHC-I (AF6-88.5.5.3), MHC-II I-A/I-E (M5/114.15.12), CD103 (2E7), CD3ε (145-2C11), CD4 (GK1.5, RM4-5), CD8α (53–6.7), CD8β (YT5156.7.7), CD69 (H1.2F3), CD44 (IM7), MERTK (108928), IFN-γ (XMG1.2), and granzyme B (GB12). Allophyocyanin-labeled H-2D$^b$ MHCI class I tetramers for IAV viral acid polymerase amino acids 224–233 (SSLENFRAYV) and WNV NS4B (2488–2496, SSVWNATTA) were obtained from the National Institutes of Health Tetramer Core Facility. Unconjugated antibodies used included IAV M2 ion channel clone 14C2(Novus Biologicals, Littleton, CO) and IFNAR1 (Leinco Technologies, MAR1-5A3). When detecting AXL expression, αAXL AF854 (R&D systems) was used with DCs and when testing AXL expression across immune cells during the course of IAV infection while αAXL C-20 (Santa Cruz Biotechnology) was used with naive alveolar macrophages. Secondary antibodies used in the study were donkey anti-mouse IgG-PE (Santa Cruz Biotechnology, clone 3744) and chicken anti-goat IgG-AF647 (Invitrogen, clone A21469). Flow cytometry data were analyzed using FlowJo software.

## Lymph node and lung single-cell suspension preparation

MLN were removed and homogenized by plunger against 40 μm strainers while suspended in complete RPMI. Lungs were minced with razor blades and placed in HBSS containing 2.5mM HEPES buffer and 1.3 mM EDTA for 37°C for 30 min while shaking. Samples were then transferred into RPMI 1640 containing 5% FBS, 2.5 mM HEPES buffer, and 0.5 mg/ml collagenase D (Roche, Indianapolis, IN) and incubated for 37°C for 60 min while shaking. Lung samples were homogenized and passed through a 40 μm cell strainer. Single cell suspensions from lung or MLN were treated with ACK lysis buffer before staining for flow cytometry.

## Intracellular cytokine staining

$2x10^6$ single-cell suspensions of lung cells were stimulated for 6 hr with phorbol 12-myristate 13-acetate (PMA, 20 ng/ml) and ionomycin (1 μg/ml) with protein secretion blocked with GolgiStop (BD Biosciences) when testing intracellular staining for IFN-γ. Staining was performed after fixing and permeabilizing cells using the BD Cytofix/Cytoperm kit (BD Biosciences) following the manufacturer's protocol.

## CD8$^+$ T cell analysis after WNV infection

Spleens of WT and $Axl^{-/-}$ mice were harvested 8 days post-infection after extensive cardiac perfusion with PBS. Splenocytes were dispersed into single cell suspensions with a cell strainer. Intracellular IFN-γ or TNF-α staining was performed after ex vivo restimulation with a D$^b$-restricted NS4B immunodominant peptide using 1 μM of peptide and 5 μg/ml of brefeldin A (Sigma, St Louis, MO). Intracellular granzyme B staining was performed in separate samples that also were stained with the APC-conjugated immunodominant NS4B tetramer.

## RT-qPCR

RNA was collected at the indicated time points and according to RNeasy mini kit (QIAGEN, Valencia, CA) manufacturer's instructions. iScript cDNA Synthesis kit (BIO-RAD, Hercules, CA) was used for reverse transcription and KAPA SYBR Fast qPCR kit (KAPA BIOSYSTEMS, Wilmington, MA) was used for qPCR reactions. Dissociation curves were used to assess specificity of products. Primers used in this study are listed in *Supplementary file 1*.

## ELISA

IFN-β was measured using VeriKine Mouse Interferon Beta ELISA kits (Pestka Biomedical Labs, Piscataway, NJ). IL-1β was detected by ELISA Ready-Set-Go (eBioscience, USA).

## Statistical analysis

Data are represented as mean ± SEM. Differences between the means of experimental groups were analyzed with two-tailed Student's *t*-test (GraphPad Software Inc., La Jolla, CA). Survival was

calculated using Kaplan-Meier plot, and survival curves were compared by the Mantel-Cox log-rank test (GraphPad Software Inc.). p values $\leq 0.05$ were considered significant.

## Acknowledgements

We thank members of the Rothlin and Iwasaki labs for discussion and advice. We thank Adolfo García-Sastre for gifting PR8 NS1-GFP to AI.

## Additional information

### Funding

| Funder | Grant reference number | Author |
| --- | --- | --- |
| National Institutes of Health | R01 AI089824 | Carla V Rothlin |
| National Institutes of Health | R01 AI081884 | Akiko Iwasaki |
| National Institutes of Health | AI064705 | Akiko Iwasaki |
| Howard Hughes Medical Institute | | Akiko Iwasaki |
| National Science Foundation | DGE-1122492 | Edward T Schmid |
| National Institutes of Health | R01 AI101400 | Michael S Diamond |
| National Institutes of Health | T32 AR007279 | Jonathan J Miner |
| National Institutes of Health | T32 AI007019 | Edward T Schmid Iris K Pang |
| American Italian Cancer Foundation | | Lidia Bosurgi |

The funders had no role in study design, data collection and interpretation, or the decision to submit the work for publication.

### Author contributions

ETS, Conception and design, Acquisition of data, Analysis and interpretation of data, Drafting or revising the article; IKP, EACS, LB, Conception and design, Acquisition of data, Analysis and interpretation of data; JJM, Acquisition of data, Analysis and interpretation of data; MSD, Analysis and interpretation of data, Drafting or revising the article; AI, CVR, Conception and design, Analysis and interpretation of data, Drafting or revising the article

### Author ORCIDs

Akiko Iwasaki, http://orcid.org/0000-0002-7824-9856
Carla V Rothlin, http://orcid.org/0000-0002-5693-5572

### Ethics

Animal experimentation: All animal procedures were approved by the Institutional Animal Care and Use Committee (IACUC) at Yale University (2015-11312) and Washington University School of Medicine (A3381-01).

## Additional files

### Supplementary files

- Supplementary file 1. Primer sequences for the indicated genes. F = forward, R = reverse.

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
