## [Decision Letter]

Thank you for submitting your work entitled "AXL is required for T cell priming and antiviral immunity" for consideration by *eLife*. Your article has been favorably evaluated by Tadatsugu Taniguchi (Senior editor) and three reviewers, one of whom is a member of our Board of Reviewing Editors.

The reviewers have discussed the reviews with one another and the Reviewing Editor has drafted this decision to help you prepare a revised submission.

In this manuscript, Schmid et al. explored the role of the *Axl* receptor tyrosine kinase in antiviral immunity in vivo. Prior studies have found that *Axl* is an important negative regulator of the type I interferon response to viruses, and that *Axl*-deficient cells are resistant to virus infection in tissue culture. In this study, the authors present the unexpected finding that mice deficient in *Axl* are more susceptible to infection with both Influenza A virus and West Nile virus, despite the fact that the DCs from these mice are more resistant to virus infection and produce more type I IFNs in in culture. Mice with conditional *Axl* deletion in dendritic cells are similarly more susceptible than WT mice. The authors find defective adaptive immune responses in *Axl^-/-^* mice, correlated with decreased production of IL-1b, a cytokine that is known to be important for T cell responses to IAV and WNV. Lastly, intranasal administration of recombinant IL-1b improved T cell responses to IAV.

While these findings are interesting and potentially important, there are significant weaknesses especially concerning the authors' claim that *Axl* in DCs are responsible for the antiviral effects observed in the mice. The authors should address the following concerns before the paper can be further considered.

1) *Axl* expression is very high on lung alveolar macrophages. In fact, lung AMs and splenic red pulp macrophages have the highest *Axl* expression (at baseline) on all immune cells. These data from Claudia Jackubzik in Immgen (immgen.org, seach for '*Axl*' and then in 'MFs, monocytes, neutrophils’). Thus, there are highly *Axl^+^* cells in the lung that the authors ignored; these cells are known to be vital in influenza immunity.

2) All lung alveolar macrophages are CD11c^+^, and this is the basis by which these cells are sorted by most labs (e.g. Hibbs, Hussell, the Immgen protocol, etc.). Thus, the authors used a Cre deleter that ablated *Axl* expression (probably) in lung AMs but chose to focus on lung DCs. There is no question lungs DCs are important, but it is conceivable their contribution might be less than lung AMs in this setting.

3) Neither the expression of *Axl* nor the deletion of *Axl* in the CD11c-Cre mice were accurately investigated at baseline, or through the time course of influenza infection. All immune cell types need to be quantified. The dynamics of *Axl* expression in the lung were not measured. The deletion measurements in brain and heart are irrelevant.

4) The interpretation of the IL-1 expressions cannot be interpreted based on the experimental set-up. The mechanism of susceptibility of the *Axl* KO, or the CD11c *Axl* KO may be unrelated to the T cell response, which could be a secondary consequence of effects on other *Axl^+^* cells in the lung.

5) In Figure 3, it will be important to include CD11c-Cre^+^, *Axl^+/+^* mice to control for potential Cre toxicity in dendritic cells as a contributor to the phenotype. Given that *Axl* is so broadly expressed, it is inaccurate to conclude that the effects are all DC-specific.

6) In Figure 7, the authors suggest that in vivo administration of recombinant IL-1b confers protection to IAV. What is the survival curve of the treated mice relative to untreated? And how does this impact antigen-specific T cell numbers and IFN-g production, as assessed in Figure 4? The global up regulation of CD44 expression in CD4 and CD8 T cells in Figure 7/F could be caused by generic inflammation rather than antigen-specific T cells.

7) The impairment of antiviral responses in *Axl^-/-^* mice could be due to the defect of *Axl^-/-^* DCs to take up virus infected cells, or alternatively the enhanced and chronic production of type-I interferons, which have been reported to compromise the production of type-II interferons (IFN-g). To distinguish these possibilities, it will be interesting to test if neutralizing antibodies against IFNAR could restore T cell function and antiviral defense in *Axl^-/-^* mice.

8) In the West Nile virus experiments (Figure 5), did the *Axl^-/-^* mice elevated viral titers and reduced survival?

---

## [Author Response]

*In this manuscript, Schmid et al. explored the role of the Axl receptor tyrosine kinase in antiviral immunity in vivo. Prior studies have found that Axl is an important negative regulator of the type I interferon response to viruses, and that Axl-deficient cells are resistant to virus infection in tissue culture. In this study, the authors present the unexpected finding that mice deficient in Axl are more susceptible to infection with both Influenza A virus and West Nile virus, despite the fact that the DCs from these mice are more resistant to virus infection and produce more type I IFNs in in culture. Mice with conditional Axl deletion in dendritic cells are similarly more susceptible than WT mice. The authors find defective adaptive immune responses in Axl^-/-^ mice, correlated with decreased production of IL-1b, a cytokine that is known to be important for T cell responses to IAV and WNV. Lastly, intranasal administration of recombinant IL-1b improved T cell responses to IAV.*

*While these findings are interesting and potentially important, there are significant weaknesses especially concerning the authors' claim that Axl in DCs are responsible for the antiviral effects observed in the mice. The authors should address the following concerns before the paper can be further considered.*

*1) Axl expression is very high on lung alveolar macrophages. In fact, lung AMs and splenic red pulp macrophages have the highest Axl expression (at baseline) on all immune cells. These data from Claudia Jackubzik in Immgen (immgen.org, seach for 'Axl' and then in 'MFs, monocytes, neutrophils’). Thus, there are highly Axl^+^ cells in the lung that the authors ignored; these cells are known to be vital in influenza immunity.*

2) All lung alveolar macrophages are CD11c^+^, and this is the basis by which these cells are sorted by most labs (e.g. Hibbs, Hussell, the Immgen protocol, etc.). Thus, the authors used a Cre deleter that ablated Axl expression (probably) in lung AMs but chose to focus on lung DCs. There is no question lungs DCs are important, but it is conceivable their contribution might be less than lung AMs in this setting.

We agree with the reviewers on these two important and interrelated points. We have now explored the function of AXL in alveolar macrophages in the response to influenza infection.

First, we tested the susceptibility of *Axl*^-/-^ alveolar macrophages to influenza infection and compared the percentage of GFP positive alveolar macrophages in WT and *Axl*^-/-^ mice infected with PR8-GFP. *Axl*^-/-^ and WT alveolar macrophages were equally susceptible to infection by PR8-GFP (Figure 2). This is in marked contrast with the increase resistance to infection in CD11c^+^CD11b^+^CD103^-^ and CD11c^+^CD11b^-^CD103^+^ DCs from *Axl*^-/-^ mice in comparison to WT mice (Figure 2).

It is interesting to note that alveolar macrophages express both AXL and MERTK (Fujimori et al., Mucosal Immunol, 2015 and Figure 2—figure supplement 1), while AXL but not MERTK is detected in both DC subsets (Figure 2—figure supplement 1). Thus, it is possible that MERTK compensates for the loss of AXL in alveolar macrophages, that MERTK is the relevant TAM receptor in these cells, or that neither AXL nor MERTK regulate the susceptibility of alveolar macrophages to PR8-GFP.

To further address the functional role of AXL in alveolar macrophages, we generated *Csf1r*-Cre^+^*Axl^fl/fl^* mice and compared their susceptibility to influenza infection with *Cd11c*-Cre^+^
*Axl^fl/fl^* mice and respective controls. While AXL is ablated in DCs and alveolar macrophages using the *Cd11c*-Cre driver (as predicted by the reviewers, Figure 4—figure supplement 2), *Csf1r*-Cre preferentially induces the ablation of AXL in alveolar macrophages (Figure 4—figure supplement 2). In the original submission we reported that *Cd11c*-Cre^+^
*Axl^fl/fl^* mice were more susceptible to succumbing to influenza infection than control mice (Figure 4). This associated with a failure to mount an appropriate antiviral T cell response (Figure 5). In contrast, *Csf1r*-Cre^+^
*Axl^fl/fl^* mice were equally susceptible to influenza infection as control mice (Figure 4). Furthermore, *Csf1r*-Cre^+^
*Axl^fl/fl^* infected mice were able to mount CD8 and CD4 T cell responses (Figure 5).

Thus, we can conclude that ablation of AXL in alveolar macrophages is not sufficient to impair the induction of antiviral adaptive immunity and confer susceptibility to influenza infection. AXL and the related RTK MERTK are not only inhibitors of the immune response but also mediate the phagocytosis of apoptotic cells. It is possible that AXL and MERTK function in alveolar macrophages is dispensable for the regulation of the priming of adaptive immunity, but participates in the clearance of apoptotic debris during the resolution phase.

We should also note that the available conditional KO lines cannot result in the selective ablation of *Axl* in DCs. As such, we cannot exclude the possibility that it is the concomitant ablation of *Axl* in DCs and alveolar macrophages, rather than the selective loss of AXL in DCs, that accounts for the increased susceptibility to infection in *Cd11c*-Cre^+^
*Axl^fl/fl^* mice. As more selective DC Cre drivers become available, we will be able to test if ablation of *Axl* solely in DCs results in impaired induction of the antiviral T cell response. We have discussed the implications of these results and revised our conclusions in light of these new findings in the revised manuscript.

3) Neither the expression of Axl nor the deletion of Axl in the CD11c-Cre mice were accurately investigated at baseline, or through the time course of influenza infection. All immune cell types need to be quantified. The dynamics of Axl expression in the lung were not measured. The deletion measurements in brain and heart are irrelevant.

We agree with this comment and have characterized the expression of AXL in CD103^+^ DCs, CD11b^+^ DCs, neutrophils, NK cells, CD4^+^ T cells, CD8^+^ T cells, B cells, and alveolar macrophages at baseline and during the course of influenza infection. These new data are included in Figure 4—figure supplement 1.The measurements in the brain and heart have been removed. The characterization on AXL deletion in the conditional KOs is included in Figure 4—figure supplement 2.

4) The interpretation of the IL-1 expressions cannot be interpreted based on the experimental set-up. The mechanism of susceptibility of the Axl KO, or the CD11c Axl KO may be unrelated to the T cell response, which could be a secondary consequence of effects on other Axl^+^ cells in the lung.

We thank the reviewers for this comment. As indicated by the reviewers in point 1 above, AXL expression in not limited to DCs and loss of AXL in other cells could account for the T cell phenotype and the increased susceptibility to influenza that we describe. One of the other main AXL expressing cells in the response to influenza, beyond DCs, are alveolar macrophages. Importantly, the use of *Cd11c*-Cre^+^
*Axl^fl/fl^* presented in our initial results did not allow us to discriminate the contributions of AXL in DCs versus alveolar macrophages, as both cell types are targeted in this mouse line.

To test if genetic ablation of AXL in alveolar macrophages could be related to the impaired antiviral T cell response, we generated *Csf1r*-Cre^+^
*Axl^fl/fl^* mice. As described above, these mice were able to mount antiviral T cell responses and survive the infection (Figure 4 and Figure 5). Thus, we conclude that ablation of AXL in alveolar macrophages is not sufficient to impair the induction of antiviral adaptive immunity. However, we cannot exclude the possibility that the ablation of *Axl* in alveolar macrophages cooperates with the loss of AXL in DCs in *Cd11c*-Cre^+^
*Axl^fl/fl^* mice to limit the induction of antiviral T cells. As discussed above, as more selective DC Cre drivers become available, we will be able to test if ablation of *Axl* solely in DCs results in impaired induction of antiviral T cell response. A discussion on these findings has been included in the revised manuscript.

5) In Figure 3, it will be important to include CD11c-Cre^+^, Axl^+/+^ mice to control for potential Cre toxicity in dendritic cells as a contributor to the phenotype. Given that Axl is so broadly expressed, it is inaccurate to conclude that the effects are all DC-specific.

This is indeed an important control. We detected no differences in viral titers between WT, *Cd11c*-Cre^+^
*Axl*^+/+^, and *Cd11c*-Cre^-^
*Axl*^fl/fl^ mice, while, as expected, *Axl*^-/-^ mice had a higher viral load. This new data is presented in Figure 4—figure supplement 4.

6) In Figure 7, the authors suggest that in vivo administration of recombinant IL-1b confers protection to IAV. What is the survival curve of the treated mice relative to untreated? And how does this impact antigen-specific T cell numbers and IFN-g production, as assessed in Figure 4? The global up regulation of CD44 expression in CD4 and CD8 T cells in Figure 7/F could be caused by generic inflammation rather than antigen-specific T cells.

We agree with the reviewers that it was important to expand on this data. We found that administration of recombinant IL-1β conferred protection to *Axl*^-/-^ mice and rescued their survival to that of WT controls (Figure 8). Administration of IL-1β to *Axl*^-/-^ mice also resulted in the recovery of IFNγ-producing antigen-specific (PA) CD8^+^ T cells as well as in IFNγ^+^CD4^+^ T cells (Figure 8).

7) The impairment of antiviral responses in Axl^-/-^ mice could be due to the defect of Axl^-/-^ DCs to take up virus infected cells, or alternatively the enhanced and chronic production of type-I interferons, which have been reported to compromise the production of type-II interferons (IFN-g). To distinguish these possibilities, it will be interesting to test if neutralizing antibodies against IFNAR could restore T cell function and antiviral defense in Axl^-/-^ mice.

We thank the reviewers for this very insightful suggestion. We tested the effect of blocking IFNAR in infected WT and *Axl*^-/-^ mice by administering MAR1-5A3 anti-IFNAR antibody. Remarkably, *Axl*^-/-^ mice treated with MAR1-5A3 were able to survive the infection similar to WT mice (Figure 7). This increase in survival correlated with a restored T cell function, as shown by the number of IFN-γ^+^ PA-restricted CD8^+^ T cells and IFN-γ producing CD4^+^ T cells (Figure 7). Similarly, the number of CD8^+^CD44^+^ and CD4^+^CD44^+^ T cells in infected *Axl*^-/-^ mice treated with the anti-IFNAR antibody were restored to that of control mice (Figure 7). In summary, these results demonstrate that the enhanced production of type I IFNs upon genetic ablation of *Axl* results in impaired induction of the protective antiviral adaptive immunity.

8) In the West Nile virus experiments (Figure 5), did the Axl^-/-^ mice elevated viral titers and reduced survival?

We have recently reported that *Axl*^-/-^ mice infected with West Nile virus (WNV) exhibited reduced survival (~80% mortality compared to ~30% mortality in WT control animals) as well as a >1-log increase in viral burden in the CNS at days 6 and 8 after infection (Miner et al., Nat Med 2015). CD8^+^ T cells play a critical role in clearing WNV from the CNS. Indeed, mice lacking CD8^+^ T cells exhibit ~95% mortality during WNV infection due to delayed clearance of WNV-infected cells from the CNS (Shrestha and Diamond, J. Virol. 2004). Thus, the deficient CD8^+^ T cell responses in *Axl*^-/-^ mice infected with WNV are likely to explain their increased mortality and viral burden. A reference to the previously reported increased susceptibility of *Axl*^-/-^ mice to WNV has been included in the revised manuscript.